# Molecular Mechanisms of Sex-Related Differences in Arthritis and Associated Pain

**DOI:** 10.3390/ijms21217938

**Published:** 2020-10-26

**Authors:** Ju-Ryoung Kim, Hyun Ah Kim

**Affiliations:** 1Division of Rheumatology, Department of Internal Medicine, Hallym University Sacred Heart Hospital, Gyeonggi 14068, Korea; jurykim75@gmail.com; 2Institute for Skeletal Aging, Hallym University, Chuncheon 24252, Korea

**Keywords:** sex difference, rheumatoid, osteoarthritis, arthritis pain, animal model

## Abstract

Clinical conditions leading to chronic pain show important sex-related differences in the prevalence, severity, and degree of functional disability. Decades of epidemiological and clinical studies have demonstrated that women are more sensitive to pain than men. Arthritis, including rheumatoid arthritis (RA) and osteoarthritis (OA), is much more prevalent in females and accounts for the majority of pain arising from musculoskeletal conditions. It is therefore important to understand the mechanisms governing sex-dependent differences in chronic pain, including arthritis pain. However, research into the mechanisms underlying the sex-related differences in arthritis-induced pain is still in its infancy due to the bias in biomedical research performed largely in male subjects and animals. In this review, we discuss current advances in both clinical and preclinical research regarding sex-related differences in the development or severity of arthritis and associated pain. In addition, sex-related differences in biological and molecular mechanisms underlying the pathogenesis of arthritis pain, elucidated based on clinical and preclinical findings, are reviewed.

## 1. Introduction

Pain is a distressing experience associated with actual or potential tissue damage, which is composed of sensory, emotional, cognitive, and social components [1]. Acute pain is provoked by a distinct injury and is usually self-limited, while chronic pain lasts longer, often persisting beyond normal tissue healing time. Pain from a wide range of diseases, including arthritis and low back problems, accounts for a tremendous burden of disability worldwide. However, pain in the context of injury is also essential for organismal survival, such that patients with congenital insensitivity to pain often fail to defend themselves against serious harm from the environment, resulting in permanent damage and even death. An understanding of the complexity of the mechanisms underlying the pathogenesis of pain in different clinical settings is therefore of the utmost importance. One area of pain research that requires clarification is sex-related difference. It has been widely reported that the prevalence rates of many different pain conditions are higher in women than in men. In addition, pain is usually more severe for women than men with the same disease condition. For example, in a population-based study of osteoarthritis (OA) in Johnston Country, North Carolina, the prevalence of radiographic knee OA was 30.8% higher among females than among males, while the prevalence of symptomatic knee OA was 38.5% higher among females [2]. This tendency of higher symptomatic than radiographic OA among females was also observed in Japanese and Korean cohorts [3]. However, research regarding the mechanisms underlying the sex-related differences in pain is still in its infancy due to the bias in biomedical research, which has largely been performed in males. In particular, preclinical studies have failed to use appropriate numbers of female animals, such that 79% of studies published between 1995 and 2005 in a journal dedicated to pain research used exclusively male rodents [4]. In addition, the sex of subjects was omitted in 22–42% of articles in various domains of research, including neuroscience [5]. Following the implementation of the 1990 policy requiring that women be appropriately represented in biomedical research studies, the National Institutes of Health (NIH) released a guide notice, “Consideration of Sex as a Biological Variable in NIH-funded Research,” in 2015, which sets forth the expectation that sex will be factored into research design, analysis, and reporting in preclinical studies of vertebrates.

In this review, current advances in clinical research regarding sex-related differences in pain (especially arthritis pain) are discussed. In addition, the biological and molecular mechanisms underlying its pathogenesis, elucidated by clinical and preclinical studies, are reviewed.

## 2. Clinical Evidence of Sex-Related Differences in Pain

### 2.1. Gender vs. Sex

Although both terms have long been used interchangeably in the scientific literature, recent reports view these two terms as conceptually distinct. “Sex” refers to anatomical, genetic, or physiological differences between males and females, and is categorized dependent upon the reproductive system, chromosomes, and the types of gametes produced [6]. On the other hand, “gender” refers to how a person views how they fit (or do not fit) into society’s expectations and gender roles based on biological sex [7]. On the whole, biomedical research still tends to use the two terms interchangeably, and lacks clarity in the application of terminology related to “sex” and “gender” [8]. For example, a retrospective cohort study of administrative data used the term “gender” while results were presented as male and female [9]. Previous databases used in biomedical research have predominantly used “sex” instead of “gender” in their variable classification. In addition, it is not yet known whether it is appropriate to classify experimental animals according to anatomy or gender. Therefore, in this article, the term “sex” is used, unless the relevant literature defines and uses “gender” according to the current scientific criteria.

### 2.2. Sex-Related Differences in Clinical Pain Perception

In a review published in 1996, Unruh examined sex-related variations in pain conditions from 105 epidemiological studies for common recurrent pain in women and men, and an additional 13 population studies of menstrual pain. In most studies, women were found to have higher prevalence rates as well as more severe levels of headaches and migraines. In addition, women report headaches of greater frequency and longer duration than men [10]. Musculoskeletal pain in the neck, shoulders, upper limbs, and hips were more frequent in women than in men, and women had more multiple pain sites, more intense pain, and more frequent pain in the majority of epidemiological studies [10]. On the other hand, there were no significant sex-related differences in the prevalence of low back pain, which may be more strongly related to occupational factors. A review published in 2012 by Racine et al. analyzed articles published between 1998 and 2008 examining sex-related differences in the perception of laboratory-induced pain in healthy subjects [11]. In contrast to epidemiological studies suggesting higher pain sensitivity in women, the majority of the studies that measured pain intensity and unpleasantness showed no sex-related differences in non-diseased subjects [11]. For example, no clear pattern of sex-related differences were detected across various types of experimental pain, including cold pain perception, ischemic pain, muscle pain, electrical pain, and chemical pain [11]. In addition, the same authors reported that hormonal and physiological factors did not affect the differences in pain sensitivity between healthy women and men [11]. Some studies suggested that the phenomena of temporal summation of pain, allodynia, and secondary hyperalgesia could be more pronounced in females than in males, which may indicate that central sensitization is augmented in healthy females. Although many studies have examined whether endogenous pain inhibitory systems could be less efficient in females than in males, the experimental evidence in support of this suggestion is mixed and does not necessarily apply to all pain modalities. Experimental manipulation of gender roles has been shown to impact sex-related differences in pain responses, such that subjects are willing to withstand a painful task to adhere to a gender-based expectation [12]. It is notable that past history of childhood sexual abuse was found to affect pain sensitivity in females but not in males [13], suggesting that pain perception may be more influenced by individual past history in females. The authors noted that laboratory studies performed in young university students may not be extrapolated to patient populations suffering from acute or chronic pain. In addition, the lack of conceptual clarity in measuring gender has historically impeded advances in determining sex/gender relationships in pain research [14].

### 2.3. Molecular Mediators of Sex-Related Differences in Arthritis

Arthritis accounts for the majority of pain arising from the musculoskeletal condition. Rheumatoid arthritis (RA), the hallmark inflammatory and autoimmune arthritis, is typically more prevalent in women, with a female to male ratio of 3:1 [15]. It is notable that the female to male prevalence ratio decreases with age, suggesting a role of female sex hormone exposure in RA risk [16]. However, there is some debate regarding the pathogenetic mechanism underlying the sex hormone effect. While amelioration during pregnancy and flare postpartum with declining estrogen levels is common among RA patients, estrogen levels are significantly elevated in the synovial fluid in both male and female RA patients due to high aromatase activity induced by locally produced inflammatory cytokines [17,18]. The efficacy of oral contraceptives or hormone replacement therapy for reducing the severity of RA is also unclear [19]. A study of 10 postmenopausal female RA patients showed that an elevated prolactin/cortisol ratio early in the morning was accompanied by higher interleukin (IL)-1β and tumor necrosis factor-alpha (TNF)-α levels, suggesting that prolactin may be another mediator of the sex-related differences in RA [20]. In contrast to the debatable role of estrogen, androgen replacement therapy has shown modest efficacy in both male and female RA patients [21,22]. Dehydroepiandrosterone (DHEA), androstenedione, and testosterone inhibit secretion of IL-1β and TNF-α, and 5α-dihydrotestosterone (DHT) inhibits activation of the human IL-6 gene promoter stimulated by nuclear factor (NF)-κB [23,24,25]. Female RA patients have lower than normal levels of DHEA and/or DHEA sulfate, and male RA patients show a negative association between levels of serum testosterone and disease severity [26]. A minor allele of SNP rs1790834 in the cytochrome B5 type A (CYB5A) gene, which converts DHEA into the metabolite 7α hydroxy-DHEA, was shown to be associated with a reduced risk of RA in women [27]. The protective allele increased CYB5A mRNA expression and activation of steroid 17, 20-lyase activity, which is the decisive step in androgen synthesis. It was suggested that the minor allele of SNP rs1790834 may help to ensure protective androgen levels in women, in whom androgen levels are generally lower than in men.

OA is a prototypical non-inflammatory arthritis, with different mechanisms underlying its pathogenesis compared with those of RA. However, female sex is also a strong risk factor for OA. Aside from the biomechanical differences, such as knee adduction moment according to sex, the influence of sex hormones on OA has been studied to elucidate the mechanisms underlying the sex-related differences. However, conflicting results have been reported, with a cross-sectional study showing a positive association between estradiol level and radiographic knee OA, while another cohort study showed a negative association [28,29]. No associations of DHEA sulfate, androstenedione, and testosterone with tibial and patellar cartilage loss or knee arthroplasty for OA were reported [30,31]. Although menopause is associated with an increase in the prevalence of OA, data regarding the influence of postmenopausal hormone replacement therapy on the development of OA are inconsistent. Data from the Women’s Health Initiative placebo-controlled, double-blind, randomized trial showed that in the estrogen-alone trial, women receiving hormone therapy had significantly lower rates of any arthroplasty, while in the estrogen-plus-progestin trial, there were no associations with total arthroplasty or individual hip or knee arthroplasties [32]. In a 4-year randomized, double-blind, placebo-controlled trial of estrogen plus medroxyprogesterone acetate, no significant effects of hormone replacement therapy on knee pain and related disability were observed [33]. Although OA is considered a non-inflammatory arthritis, some studies show a relationship between pro-inflammatory cytokines and sex difference. In a study of symptomatic knee OA, serum IL-6 was associated with increased knee bone marrow lesions (BMLs) in both females and males, while serum IL-17F and IL-23 predicted increased knee BML scores in females only, suggesting that inflammation is involved in BML pathogenesis in knee OA, especially in women [34].

### 2.4. Molecular Mediators of Sex-Related Differences in Arthritis Pain

Arthritis is a good disease model for elucidation of the mechanisms underlying the pathogenesis of chronic pain and its sex-related differences. A meta-analysis of 16 RA cohort studies including 21,612 females and 6871 males examined sex-related differences in pain, and showed that the standardized mean difference in pain visual analog scale (VAS) score was significantly higher in females [35].

For OA, women reported higher ratings of pain than men in studies performed in a variety of regions and ethnic groups [36,37,38]. Sex-related differences were not found to result from more severe OA in women [39]. Although psychological factors, including coping with pain, mood, and sensitivity to pain, may play roles in mediating the sex-related differences in arthritis pain, there is a paucity of data regarding the molecular aspects underlying the differences in pain mechanisms. The amygdala, which plays an important role in the emotional-effective dimension of pain as well as cognitive aspects such as pain-related decision-making through interactions with cortical areas, may also contribute to sex difference in pain and is an area of future research. In a recent report, increased C-reactive protein (CRP) levels were shown to be associated with greater painful joint count in OA among women, but not among men, suggesting a sex-specific association of acute-phase reaction and OA pain [40]. Sex-related differences in body fat mass may be important because obesity is a strong risk factor for both OA and pain. Interestingly, total fat mass and fat/muscle mass ratio were significantly and positively associated with musculoskeletal pain only among female subjects in both cross-sectional and prospective analyses [40,41]. In addition, increased fat/muscle ratio was significantly associated with disease activity score (DAS) 28-P, which is a derived index from DAS 28 to assess the contributions of noninflammatory factors to pain, only [42] in female RA patients. As fat tissue is considered to be an endocrine organ producing proinflammatory adipokines, the stronger inflammatory response arising from increased fat in women may play a role in the sex-related differences in pain [43]. A study of female knee OA patients showed that levels of synovial fluid adiponectin were positively correlated with pain, whereas resistin and visfatin showed significant positive and negative associations with disability, respectively, after adjustment of potential confounders [44]. Plasma leptin and adiponectin levels were positively, while adipsin was negatively, associated with regional symptomatic joint count in female OA patients, while resistin showed a negative association in men [45]. Although a meta-analysis showed that serum leptin levels were higher in RA patients with high disease activity, another study showed that it was not correlated with the number of painful joints in RA patients [46,47].

Sex hormones may account for the sex-specific association between fat mass and pain. There are marked sex-related differences in fat mass parameters, such as a lower amount of visceral adipose tissue, greater lower body fat stores, as well as higher fat percentage, in women compared to men [48]. While the actions of androgens in white adipose tissue mostly explains sex-related differences in body fat distribution, estrogen also has an influence on fat accumulation, such that estrogen deficiency after menopause decreases fat oxidation and increases subcutaneous adipose tissue storage of free fatty acid [49], leading to a postmenopausal increase in fat mass. This suggests that estrogen deficiency may mediate the relationship between pain and increased fat mass after menopause.

Despite a number of studies reporting sex-related differences in response to analgesics, there have been few studies of this aspect in relation to arthritis pain. The multinational, observational Measurement of Efficacy of Treatment in the Era of Outcome in Rheumatology (METEOR) register study showed that there were no differences in response to treatment between men and women with RA; however, separate analysis of pain responses between the sexes was not reported [50]. In a study of 340 patients with axial spondyloarthritis (SpA) who were treated with anti-TNF-α drugs, female sex was associated with lower rates of response to treatment and of disease remission [51]. However, as in RA, a composite index for assessment of response was used, and pain response data were not available. A study exploring sex-related differences in placebo response to analgesic medication in knee OA patients showed that, although placebo effects emerged in both sexes, women showed greater placebo response represented by 6-min treadmill distance [52]. No consistent treatment-by-sex interaction was observed for knee OA patients taking rofecoxib, which showed consistent efficacy in both sexes [53]. On the other hand, women with knee OA showed a significantly lower response to treatment than men for intra-articular injections of sodium hyaluronate and corticosteroid with regard to pain relief [54]. Table 1 briefly summarizes the sex-related differences in human arthritis and pain.

## 3. Animal Study Evidence of Sex-Related Differences in Pain

### 3.1. Molecular Mechanism of Sex-Related Differences in Animal Pain Behavior

Studies in animal models to understand how pathogenesis of pain and analgesic efficacy have added valuable information regarding the complex nature of pain as a human pathology [55]. Although there is a great deal of interest in sex-dependent pain pathophysiology, few studies have addressed the issue of sex-related differences in animal models, because much of what is known about the pathological mechanisms of pain were identified in male animals. In recent years, the underrepresentation of female animals not only in the field of pain research but also in the majority of other disease models has been addressed. Therefore, the need to use both sexes remains a key priority for pain investigators working with experimental animals.

Recent studies using animal models have demonstrated sex-related differences in the induction and maintenance of chronic pain, such as neuropathic, inflammatory, and arthritis pain. Sex-related differences in pain have been explored and evidence suggest an influence of sex steroid hormones on pain perception. There is a paper showing that pain stress related factors including plasma glucose, fatty acid, and corticosterone were significantly increased together with increased plasma estradiol in female estrous rats compared with male estrous rats in a formalin-induced chronic pain model. This result suggests that estradiol is related to a sensibility to pain and that the estrous cycle has an effect of sex difference modulator on pain and nociceptive sensibility [56].

Sex-related differences involving peripheral and central immune cells, such as microglia, macrophages, astrocytes, mast cells, and T cells, are essential for chronic pain hypersensitivity. Multiple lines of evidence have established that spinal microglia cells, macrophage-like immune cells that reside in the central nervous system, are reactive to peripheral inflammatory responses or nerve damage induced by spared nerve injury (SNI) and can produce mechanical hypersensitivity [57,58,59]. In both male and female mice, microglia proliferate in the spinal cord following SNI, but the involvement of microglia as mediators of persistent pain hypersensitivity has only been demonstrated in male mice [60]. As supporting evidence that microglia play a role in sex-related differences in pain, intrathecal injection of three different glial inhibitors after SNI, i.e., minocycline, fluorocitric acid, and propentofylline, resulted in dose-dependent reversal of mechanical allodynia in male mice, while no reversal of allodynia was observed in female mice, suggesting that microglia are not necessary for pain hypersensitivity in females. Furthermore, this function of microglia was confirmed by the observation that P2X purinoceptor 4 (P2X4R), which are purinergic receptors, are expressed specifically in microglia and are essential in mediating pain hypersensitivity in response to nerve injury [59]. Inhibition of spinal P2X4R reverses pain hypersensitivity in male but not female mice, as females do not show upregulation of P2X4R in spinal microglia. Interferon regulatory factor (IRF)-8 upregulates P2X4R expression on microglia after nerve injury by activating IRF5, which binds specifically to the promoter of the P2rx4 gene; however, Irf8 and Irf5 showed equivalent upregulation after SNI in both sexes [60]. Therefore, it is likely that IRF regulate P2rx4 transcription in a sex-dependent manner. Inhibition of spinal p38 MAP kinase prevents formalin-induced inflammatory pain and chronic constriction injury (CCI)-induced neuropathic pain only in male mice, but not in female mice. The p38 phosphorylation level is higher after injury in males than in females [61]. This sex-dependent alleviation of neuropathic pain by inhibition of spinal p38 MAP kinase was also observed in rats [61].

Toll-like receptor 4 (TLR4) is located primarily on microglia and has been implicated in pain pathology, with evidence that systemic and intrathecal administration of lipopolysaccharide (LPS) results in TRL4 activation leading to pain hypersensitivity in rats [62,63], and that TLR4 knockout mice exhibit decreased pain hypersensitivity after nerve injury. Interestingly, these results were found only in male animals, suggesting that pain hypersensitivity mediated by TLR4 is sex dependent. In addition, activation of TLR4 in the spinal cord with intrathecal LPS results in robust mechanical allodynia only in male mice, while LPS allodynia was not observed in female mice, indicating the existence of a TLR4-independent spinal pathway for pain processing in females [64]. The role of TLR4 in pain hypersensitivity in male mice is dependent on the male hormone testosterone, such that administration of testosterone to primary macrophages from orchiectomized (ORX) animals elicits a significant decrease in the expression of TLR4 [65]. There is accumulating evidence that cytokines may also be differentially regulated in males vs. females following peripheral nerve injury. In a recent study of neuropathic pain induced by sciatic nerve injury, females exhibited greater numbers of Th17 proinflammatory T cells and specific responses with higher levels of IL-17A compared to males at both the injured nerve and at the corresponding spinal cord level [66]. In addition, in a neuropathic pain model, females showed pain relief after treatment with β2-integrin antagonist BIRT377, via marked reduction in the expression of IL-17A in sciatic and spinal cord tissues in comparison with males. This study suggested that differences in T cell bias toward a proinflammatory state may underlie sex-related differences in neuropathic pain [66].

There is a great deal of ongoing effort to elucidate sex-related differences in gene transcription in the dorsal root ganglion (DRG) using rodent pain models. Stephens et al. performed RNA-Seq analysis in the lumbar DRG following CCI in both sexes, and both common and sex-specific gene expression following CCI were identified, including inflammatory cytokines, growth factors, and neurotransmitters. They showed that the expression level of the cytokine, colony-stimulating factor 1 (*csf1*), is 1.7 times higher in females than in males [67]. Csf1 is known to be transported from the DRG to the spinal cord where it binds to its receptor, located on microglia during neuropathic pain. As csf1 was reported to be expressed de novo in injured sensory neurons following peripheral nerve injury [68], sex-specific regulation of csf1 may lead to sex-related differences in susceptibility to neuropathic pain after nerve injury. In addition, A-kinase anchor protein 9 (*Akap9*), a molecule involved in the regulation of membrane potential, is significantly upregulated in females but not in males after CCI. It may play a role in sex-related differences in pain development by increasing neuron excitability in females following nerve injury [69]. In male mice, *Oprm1*-encoding μ-opioid receptor (MOR) is significantly upregulated compared to females. As several animal studies showed that morphine contributes to greater analgesia in male animals than in females [70,71], Oprm1 may influence sex-related differences in peripheral mechanisms of morphine analgesic efficacy and of endogenous analgesic mechanisms [72]. Other recent sex differences include the female-specific analgesic effects of pharmacological treatment in a peripheral inflammatory pain model. Cerebrolysin is a multimodal neuropeptide preparation, which can modify the neuroprophic factor to produce a neuroprotective effect, and Morales-Medina et al. has reported that cerebrolysin reversed the mechanical allodynia in females but not in males in carrageenan-treated rats [73]. Other analgesic effects were also shown in female mice, but not in male mice via administration of fluoxetine in formalin-induced chronic pain model mice [74]. Fluoxetine, one of the serotonin reuptake inhibitors, increased the expression of metabotropic glutamate receptor type 2 (mGlu2) in the dorsal horn and DRG of female mice together with a decreased expression of the epigenetic-modifying enzyme, histone deacetylase 2 (HDAC2), suggesting a molecular explanation for the analgesic effects in female mice [74]. Table 2 briefly summarizes the sex-related differences in animal pain.

### 3.2. Molecular Mechanisms of Sex-Related Differences in Animal Models of Arthritis

Animal models of RA can be roughly divided into immunization or transfer models. Collagen-induced arthritis (CIA), the most commonly used animal model of RA, is induced by intradermal injection of heterologous type II collagen (CII) in complete Freund’s adjuvant (CFA) leading to an autoimmune response in the joints [75,76]. Antigen-induced arthritis (AIA) is triggered by the injection of exogenous antigens leading to subsequent pathology, including immune complex-mediated inflammation followed by articular T cell activation [77]. The most widely used transfer model is the collagen antibody-induced arthritis (CAIA) model, which is induced by injecting a cocktail of anti-CII antibodies and LPS [78]. A convenience of this animal model for RA induction is that this model is applicable independent of mouse strain or genotype. The K/BxN serum transfer arthritis model is induced by injecting anti-glucose-6-phosphate isomerase (anti-GPI)-positive serum from K/BxN mice into commonly used mouse strains [75,79]. The sex-related differences have been confirmed in an animal experiment of RA, and most of them have been focused on the influence of sex hormones, because they function as inhibitors or suppressors of immune responses. Sex hormones contribute to the development and activity of the immune system, and both innate and adaptive immune systems bear receptors for sex hormones and respond to hormonal cues [80]. Generally, male mice are known to be more susceptible to CIA than female mice [81] and ovariectomy of female DBA/1 mice exacerbated RA induced by CIA, suggesting a role for estrogen in attenuating autoimmune arthritis. When females were treated with low doses of estrogen, the pathogenic incidence and severity of RA were significantly reduced [82]. The results of this animal study were inconsistent with human RA, in which the efficacy of estrogen has been demonstrated [18]. In contrast with mice, in a rat CIA model, more robust Th1/17 responses were observed in female rats and females exhibited a higher incidence of arthritis compared to their male counterparts [83]. In a CAIA animal model, male mice showed an increased incidence of arthritis compared to female mice before LPS injection; however, this sex-related difference was abolished after LPS injection. When male CAIA mice were treated with 17β-estradiol (E2), it led to less severe disease with no influence on arthritis development [84]. A number of studies have indicated a role for cyclooxygenases (COX) in inflammatory arthritis, most of which supported an important role for COX-2, an inducible enzyme. Both COX-1 and COX-2 have an effect on inflammatory arthritis severity in a sex- dependent manner. For example, chronic Freund’s adjuvant-induced arthritis in COX-1^−/−^ and COX-2^−/−^ showed attenuation of edema and joint destruction only in females. Notably, neither male nor female COX-2^−/−^ mice developed thermal hyperalgesia or mechanical allodynia, suggesting the importance of COX-2 in the generation of pain. Interestingly, female COX-1^−/−^ mice showed reduced contralateral allodynia compared with male COX-1^−/−^ or wild-type mice [85].

Animal models of OA include surgical destabilization of medial meniscus (DMM), meniscectomy (MNX), anterior cruciate ligament transection (ACLT)] and chemically induced models (monoiodoacetate-induced arthritis (MIA), collagenase-induced OA (CIOA)) as well as spontaneous models. Huang et al. investigated whether age affects OA progression following DMM in female or male mice [86]. Aged male mice developed more cartilage degeneration with subchondral bore changes compared to aged females. In a study of young (~4 months) mice, ORX male and ovariectomized (OVX) female mice were used to investigate the roles of sex hormones in the development of OA after DMM. Female mice were found to be less prone to developing OA following DMM compared to their male counterparts, and OA was more severe in OVX females and less severe in ORX males, suggesting that both sex hormones play critical roles in the progression of OA. Most spontaneous models of OA, whether naturally occurring or induced by generic modifications [87], show that male mice exhibit more severe OA. STR/ort male mice exhibiting spontaneous OA show a higher prevalence of OA with alteration of bone structure, including an earlier increase in angular degrees of internal tibial torsion compared to females. In IL-6-deficient mice, which also develop spontaneous OA, more extensive cartilage loss is observed in males than in females upon aging. In addition, IL-6^−/−^ males show more extensive extracellular matrix (ECM) depletion and subchondral bone sclerosis compared to females, and cartilage proteoglycan (PG) synthesis and bone mineral density are decreased to a greater extent in males than in females, suggesting a protective role of IL-6 in age-related OA in male mice [88]. Another genetically modified mouse targeted NOV (Nephroblastoma overexpressed), which is a member of the CCN family of matricellular proteins. Nov^del3−/−^ males exhibited severe OA-like pathology at 12 months, affecting all tissues of the joint, including destruction of the articular cartilage, meniscal enlargement, osteophyte formation, and expansion of fibrocartilage in comparison to females [89].

### 3.3. Molecular Mechanism of Sex-Related Differences in Animal Models of Arthritis Pain

As the majority of arthritis animal studies used male animals, the mechanisms of arthritis pain have mostly been examined only in males. RA features increased immune cell infiltration in the synovium and elevated production of various proinflammatory mediators, including cytokines. This inflammatory response is a risk factor for both joint destruction and pain. However, the degree of inflammation is not always concordant with that of pain. For example, in animal models of RA as well as in human RA patients, despite resolution of inflammation, pain may still persist, indicating that more complex mechanisms other than overt inflammation contribute to pain. Woller et al. noted that in the K/BxN serum transfer model, acute inflammation, and concurrent tactile allodynia are indistinguishable between male and female C57BL/6 mice [90]. However, in the post-inflammatory phase, a significant reversal of the tactile allodynia was shown in the female mice but not in males, suggesting underlying different systems processing nociceptive information between the sexes [90]. Both male and female mice showed increased spinal TNF-α mRNA expression at acute and intermediate phases, which declined at the chronic phase. On the other hand, an increase in spinal interferon (IFN)-β mRNA expression, which induces anti-inflammatory gene expression, in the chronic phase was observed only in females, but not in males. Coadministration of intrathecal IFN-β and anti-TNF antibodies reversed tactile allodynia, suggesting that spinal TNF-α and IFN-β may be involved in the transition from acute to chronic tactile allodynia and pertinent sex-related differences in pain behavior [90].

A study reported sexual dimorphism of pain behavior mediated by spinal pain signaling molecules in CAIA mice. The study focused on the late phase of arthritis and showed that the intensity of ionized calcium-binding adapter molecule 1 (IBA1) and glial fibrillary acidic protein (GFAP), which are markers of activated microglia and astrocyte, respectively, in the spinal cord did not differ between male and female animals [91]. However, a significant reversal of mechanical thresholds by intrathecal administration of glial inhibitors, minocycline and pentoxifylline, was observed only in male mice, suggesting microglia-dependent pain regulation in males. To investigate differences in the transcriptome of microglia between males and females in the context of RA pain, RNA-seq analysis of sorted CD45^+^, CD11b^+^ microglia from the lumbar dorsal horn of CAIA mice during the post-inflammatory phase was performed [91]. Although genome-wide RNA sequencing analysis has indicated several transcriptional differences, including *Ddx3y*, *Eif2s3y*, and *Xist*, between microglia from males and females, no convincing differences were identified between control and CAIA groups. Taken together, these findings indicate that a subtle sex-related differences seems to exist in microglial expression profiles independent of RA. A study examining the contribution of sensory neurons in the ankle joint and adjacent tissue to the development of pain in a female CIA rat model [92] showed that joint innervating neurons exhibited enhanced calcitonin gene-related peptide (CGRP) expression in the dorsal horn, and blockade of this CGRP expression attenuated established mechanical hypersensitivity, suggesting that central mechanisms play critical roles in RA-like chronic inflammatory pain. To examine whether this finding is involved in sex-related differences in analgesic efficacy of CGRP blockers, further studies regarding pain hypersensitivity should be performed in CIA model animals of both sexes.

There is a paucity of data on sex-related differences in animal models of OA pain. Temp et al. recently reported the effect of sex on pain sensitization in C57BL/6J mice with OA induced by medial meniscal transection (MMT) surgery [93]. In males, MMT triggered biphasic mechanical hypersensitivity and decreased the load on OA limbs, with acute postoperative (1–5 days) and chronic (3–12 weeks) OA phases separated by a period of remission in the intermediate phase (1–2 weeks). On the other hand, females showed a less pronounced biphasic pattern, with greater mechanical hypersensitivity, but not poorer limb use, than males in the intermediate phase. In both sexes, neither heat hypersensitivity nor changes in locomotor activity were observed in the chronic phase. Cartilage damage was more severe in males than in females, but knee damage was not correlated with pain. The molecular mechanisms underlying these sex-related differences in pain were not explored. In a study of the effects of sex and age on pain sensitivity in rats by monoiodoacetate (MIA) injection, more pronounced and longer lasting hyperalgesia was observed in older than in young rats, while greater pain responses and higher susceptibility were observed in female than in male rats [94]. These results suggested that female sex and aging are associated with an elevated pain response in OA. On the other hand, in our recent study, joint damage and pain hypersensitivity after destabilization of medial meniscus (DMM) surgery were similar in both male and female C57/BL6 mice [95]. Although DMM surgery increased the expression of transient receptor potential cation channel subfamily V member 1 (TRPV-1) in the DRG of both males and females, the analgesic effect of capsazepine (CPZ), a TRPV-1 antagonist, was observed only in male mice with reduced expression of TRPV-1 in DRG after treatment [95]. Sannajust et al. reported sex-related differences in pain using a temporomandibular joint MIA model in rats. The results showed that females developed ongoing pain at a fivefold lower concentration of MIA than males. Treatment with MIA led to the spread of tactile hypersensitivity from the face to the forepaws and hind paws, indicating the development of central sensitization in both sexes [96]. Table 3 briefly summarizes the sex-related differences in arthritis and arthritis pain in animal models.

## 4. Conclusions

Clinical conditions leading to chronic pain show important sex-related differences in both prevalence and degree of functional disability. A number of animal models mimicking human arthritis pain have been developed, and there is increasing interest in elucidating the mechanisms underlying the sex-dependent differences in arthritis pain using these animal models. This review presented distinct molecular mechanisms leading to sex-related differences in arthritis pathology and arthritis pain based on both clinical and preclinical research findings. We suggest that molecular mechanisms, including different immune cell types, TLR4-dependent/or independent pathways, and inflammatory cytokines, contribute to the sex-related differences in arthritis-induced pain in animal models. There are still not enough studies to elucidate the molecular mechanisms underlying the pathogenesis of sex-related differences in pain, even in animal models. Uncovering such underlying mechanisms would lead to more targeted therapeutic approaches based on sex in many chronic pain conditions.

## Figures and Tables

**Table 1 ijms-21-07938-t001:** Sex difference in human arthritis and pain.

Type of Disease	Sex Difference in Joint Pathology	Sex Difference in Pain	Presumptive Molecular Mediators	References
RA	- More prevalent in women with a female to male ratio of 3:1 - Amelioration of inflammation during pregnancy and flare postpartum	- Standardized mean difference in pain visual analog scale (VAS) score was significantly higher in females	- Estrogen levels are significantly elevated in the synovial fluid in both male and female RA patients - Androgen replacement therapy has shown modest efficacy in both male and female RA patients	[15,16,17,21,35]
OA	- Female sex is a strong risk factor for OA - Menopause is associated with an increase in the prevalence of OA	- Women reported higher ratings of pain than men - Women with knee OA showed significantly less pain relief than men after intra-articular injections of sodium hyaluronate and corticosteroid	- Increased C-reactive protein (CRP) levels were shown to be associated with greater painful joint count in OA among women, but not men - In female knee OA patients, levels of synovial fluid adiponectin were positively correlated with pain - Plasma leptin and adiponectin levels were positively, while adipsin was negatively, associated with regional symptomatic joint count in female OA patients, while resistin showed a negative association in men	[40,44,45,54]

Abbreviations: RA, rheumatoid arthritis; OA, osteoarthritis.

**Table 2 ijms-21-07938-t002:** Sex difference in animal pain.

Type of Pain/Type of Disease	Animal Species	Model (Trigger)	Pain Assay/Measurement	Findings	References
Inflammatory pain	Male and female Rat, Wistar	Formalin	- Biochemical parameters (plasma estradiol, glucose, fatty acid, corticosterone)	- Female estrous rats show pain stress related factors including plasma glucose, fatty acid, and corticosterone were significantly increased together with increased plasma estradiol compared to male estrous rats	[56]
Neuropathic pain	Male and female mice	SNI	- von Frey	- Inhibition of spinal P2XRs reverse mechanical hypersensitivity in males but not in females	[59]
Neuropathic pain	Male and female rats, Sprague Dawley	CCI or SNI	- von Frey	- No sex difference in microglial activation - No sex difference in mechanical hypersensitivity - Increase of P2X4R expression and activity in male but not female rats - Inhibition of spinal microglia reverse mechanical hypersensitivity in males but not in females	[60]
Inflammatory and neuropathic pain	Male and female Mice (CD-1) or Rats (Sprague Dawley)	- CCI - Formalin	- spontaneous pain behavior - von Frey	- Phosphorylation of p38 is higher after injury in males than females - Inhibition of spinal p38 MAP kinase prevents both inflammatory pain and CCI induced neuropathic pain only in male mice and rats but not in female mice and rats	[61]
Neuropathic pain	Male and female mice, CD-1	LPS	- von Frey	- TLR4 knock out males decreased pain hypersensitivity, while females do not	[64]
Neuropathic pain	Male and female mice, C57BL/6	CCI	- von Frey	- Females exhibit greater Th17, a profound pro-inflammatory T cell, and specific responses with higher levels of IL-17A compared to males - Females showed pain relief by treatment with a β2-integrin antagonist, BIRT377, as compared to males	[66]
Neuropathic pain	Male and female rats, Sprague Dawley	CCI	- RNA-seq analysis	- *Csf1* (colony-stimulating factor 1), *Akap9* (A-kinase anchor protein 9) is significantly upregulated in females compared to males - *Oprm1* is significantly upregulated in males compared to females	[67]
Inflammatory pain	Male and female Rat, Wistar	Carageenan	- von Frey	- Cerebrolysin reversed the mechanical allodynia in females but not in males	[73]
Inflammatory pain	Male and female Mice, CD-1	Formalin	- spontaneous pain behavior	- Fluoxetine, one of the serotonin reuptake inhibitors, increased the expression of metabotropic glutamate receptor type 2 (mGlu2) in the dorsal horn and DRG of female mice but not in males - Fluoxetine decreased expression of the epigenetic modifying enzyme, histone deacetylase 2 (HDAC2) in females but not in males	[74]

Abbreviations: SNI, spared nerve injury; P2XRs, purinergic receptors; CCI, chronic constriction injury; CD-1, MAP kinase; mitogen-activated protein kinase; LPS, lipopolysaccharide; TLR4, toll-like receptor 4; Th1/17 T helper (Th)1, 17; IL-17A, Interleukin 17A; BIRT377, the LFA-1 antagonist; Csf-1, colony-stimulating factor 1; Akap9, A-kinase anchor protein 9; Oprm1, a molecule encoding μ-opioid receptor.

**Table 3 ijms-21-07938-t003:** Sex difference in arthritis and arthritis pain in animal models.

Type of Pain/Type of Disease	Animal Species	Model (Trigger)	Pain Assay/Measurement	Findings	References
RA	Male and female mice, DBA/1	CIA (CII)	- Histology of joint	- Males are more susceptible to CIA than females	[81]
RA	Female mice, DBA/1	CIA	- Histology of joint	- Castration of female DBA/1 mice exaggerated RA - Females treated with low doses of β-oestradiol, showed reduced the severity of RA	[82]
RA	Male and female rats, DA	CIA (CII/adjuvant)	- Th17/Treg ratio - Th17 cell redifferentiation - Treg trans-differentiation into exTregs	- Females exhibit greater susceptibility to CIA induction that males - More robust Th1/Th17 responses were observed in females than males	[83]
RA	Male and female mice,	CAIA (Anti-CII Ab + LPS)	- Histology of paws	- Males are more susceptible to than females - The E2-treated castrated females had a clearly decreased RA related incidence - In the non-castrated male mice, the E2 treatment led to less severe disease	[84]
RA	Male and female mice,	CFA	- Hargreaves - Von frey - Inflammatory edema, - Histology of joint	- COX-1^−/−^ and COX-2^−/−^ showed reduced edema and joint destruction in females, but not in males - No sex difference in thermal hyperalgesia or mechanical allodynia in COX-2^−/−^ - Female COX-1^−/−^ mice show reduced contralateral allodynia compared to male or WT mice	[85]
RA	Male and female mice, C57BL/6 and TLR4 null	K/BxN	- von Frey - hind paw swelling	- All mice (both sex of WT and TLR4^−/−^develop an initial tactile allodynia - Female WT and both sex of TLR4^−/−^ mice partially resolve their allodynia in the post-inflammatory phase - Female TLR4^−/−^ mice began to recover earlier than male TLR4^−/−^ mice - No sex difference between strains or sexes in hind paw swelling	[90]
RA	Male and female mice, C57BL/6, Balb/c, and CBA	CAIA (Anti-CII Ab + LPS)	- von Frey - Microglia isolation - RNA-seq analysis	- No sex difference in level of microglia cell in spinal cord - Mechanical hypersensitivity is reversed by microglia inhibitors only in males - No convincing difference in gene expression between male and females in RNA-seq analysis	[91]
OA	Male and female mice, 129S6/SvEv	DMM (surgical)	- Cartilage explant - PG quantitation - Histology of cartilage	- OVX female mice had significantly more severe OA lesions than the control females - ORX males developed significantly less severe OA than control males - ORX males supplemented with exogenous DHT, the severity of OA was restored to the level of control males	[87]
OA	Male and female mice, IL-6^−/−^ null	Spontaneous	- Histology of cartilage - PG density - knee BMD	- IL-6^−/−^ males developed more severe spontaneous OA than females - IL-6^−/−^ males showed more extensive ECM deposition and subchondral bone sclerosis than females - IL-6^−/−^ males showed reduced cartilage PG synthesis and BMD than females	[88]
OA	Male and female mice, *Nov^del3^* null	Spontaneous	- X-ray - Histology of cartilage	- Nov^del3−/−^ males exhibited severe OA-like pathology including cartilage destruction, meniscal enlargement, osteophyte, and expansion of fibrocartilage compared to females at 12 months - Subchondral sclerosis, changes in ECM composition and proliferating articular cell was greater in Nov^del3−/−^ males that females	[89]
OA	Male and female mice, C57BL/6 and	MMT (Surgical)	- Dynamic weight bearing - von Frey - locomotor activity - Hargreaves - Histology of cartilage	- No difference in heat sensitivity between sexes in acute phase but more delayed intermediated phase in females than males - Females showed greater mechanical hypersensitivity and locomotor activity in acute phase than males - Male exhibited more severe cartilage destruction than females - Load on limb decreased in males than females	[93]
OA	Male and female rats, Fischer 344	MIA (Chemical)	- Hargreaves - Pressure - Nocifensive behaviors by capsaicin	- Old rats lasted longer with hyperalgesia than young rats - Reduction in weight-bearing response was greater in old rats than in young rats - Females showed greater thermal hypersensitivity and higher susceptibility to OA than males - No sex difference in capsaicin induced nocifensive response	[94]

Abbreviations: DBA/1, RA, rheumatoid arthritis; CIA, collagen-induced arthritis; CII, collagen type II; Th17/Treg, T helper 17/regulatory T cell; Th1/17, T helper Th1/17; CAIA, collagen antibody-induced arthritis; E2, 17β-estradiol; CFA, complete Freund’s adjuvant; COX-1 and 2, cyclooxygenases-1 and -1; TLR4, toll-like receptor 4; OA, osteoarthritis; DMM, destabilization of medial meniscus; OVX, ovariectomize; ORX, orchiectomized; DHT, *dihydrotestosterone*; *IL-6*, *interleukin-6*; *ECM*, *extracellular matrix*; *PG*, *proteoglycan*; *BMD*, *bone mineral density*; Nov^del3^ CCN family of matricellular proteins, MMT, medial meniscal transection; MIA, monosodium iodoacetate.

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
