# Peer review of "Molecular Mechanisms of Sex-Related Differences in Arthritis and Associated Pain"

_ijms, 2020, doi:10.3390/ijms21217938_

Round 1
Reviewer 1 Report
Currently, there are several reviews in sex-difference including pain.
This review needs to be more focus on arthritis pain.
The section titled: Animal study evidence of sex-related differences in pain does no add any additional information. Also, this review did not compare the mechanism of arthritis pain vs other pain models. So, the rationale of have this section is not clear.
Sex-difference is the key component of this review, yet no attempt to address estrous-cycle and arthritis pain is noted.
Brain areas such as amygdala has been shown to play a key role in the arthritis pain. This need to discussed and as well as how this could play a role in sex-difference in pain.
Other mechanism(s) implicated in pain control have not be discussed in this review. This need to be discussed, and how this can play a role in sex-difference
The tables need to be improved.
Author Response
To
Reviwer#1
We appreciate the reviewer’s comments for improving our manuscript. Chronic and arthritis pain have been extensively investigated, however , so far, the study about sex difference in these pain s has not been sufficient. I would like the reviewer to understand this point.
This review needs to be more focus on arthritis pain.
➔ 1)To focus on arthritis pain, some of the contents less relevant for arthritis pain was removed (in the part of 2.1 gender vs. sex). 2) sex differences in analgesic effects of pharmacological treatment for peripheral inflammatory pain model was added. Please see the highlight in yellow on line 346~355.- The section titled: Animal study evidence of sex related differences in pain does no add any additional information.
➔ Because the main theme of this paper is on pain associated with arthritis, and the addition of other pain model such as neuropathic pain would need a whole new manuscript, such comparison was not done. - Sex difference is the key component of this review, yet no attempt to address estrous cycle and arthritis pain is noted.
➔ Although there are some papers explaining the association of estrous cycle and arthritis pain, few papers have studied sex difference in the relationship. I referenced one paper and added sex difference in pain stress related factors between female estrous rats and male estrous rats in formalin induced chronic pain model . Please see the highlight in yellow on line 280~286. - Brain areas such as amygdala has been shown to play a key role in the arthritis pain. This need to be discussed and as well as how this could play a role in sex difference in pain.
➔ Although I spent a lot of time on searching appropriate papers to answer this question, unfortunately , there’s no report showing sex difference in amygdala in the arthritis pain. But I agree with you. A mygdala may contribute in pain. Please see the highlight in yellow on line 157--160. - Other mechanism(s) implicated in pain control have not be discussed in this review. This need to be discussed, and how this can play a role in sex--difference.difference. ➔ As I answered to the question 1, I added sex differences in analgesic effects of pharmacological treatment for peripheral inflammatory pain model. Please see the highlight in yellow on line 346~355.
- The tables need to be improved.The tables need to be improved.
➔ I added some contents in the table 1. Please see the highlight in yellow.
Reviewer 2 Report
Ju-Ryoung Kim and Hyun Ah Kim have highlighted sex-related differences on the development and severity of arthritis and pain in the clinical and preclinical set-up in this review. Further, the molecular mechanisms underlying sex-related differences are emphasized. Generally, authors have touched an important subject which is underrepresented, not only in the field of arthritis and inflammatory pain but in other research areas. However, there are some points that must be reviewed.
- Title need to be revised with the removal of repetitive words. “Molecular mechanisms of sex-related differences in arthritis and associated pain” can be enough.
- Tables based on human studies similar to ones with animal studies will make it easier for readers to follow the review and understand and compare results of different studies.
- Line 135 “Osteoarthritis (OA) is a prototypical non-inflammatory arthritis,…”, however, in light of numerous recent studies, authors should also mention the inflammatory component in OA.
- TNF-α should be replaced by only TNF.
- Authors should be careful about using the term “Animal pain” (line 205), it is more appropriate to use “animal pain behavior”.
- References 7, 89 need to be corrected
- Authors should be uniform in using words “arthritis pain” or “arthritic pain”
- Line 352 reference is missing
- Line 280, “animal arthritis” to “animal models of arthritis”
- Line 336, “animal arthritis pain” to “animal models of arthritis pain”
- Line 206,” experimental animal models” should be replaced either by “animal models” or “experimental models”
- Line 335, Fibrocartilage, link should be removed.
Author Response
To Reviwer# 2
We appreciate the reviewer’s comments for improving our manuscript.
1. Title need to be revised with the removal of repetitive words. “Molecular mechanisms of sex-related differences in arthritis and associated pain” can be enough.
➔ The title was change as you correct me.
2. Tables based on human studies similar to ones with animal studies will make it easier for readers to follow the review and understand and compare results of different studies.
➔ We added a table about sex difference in human arthritis and pain (Table 1)
3. Line 135 “Osteoarthritis (OA) is a prototypical non-inflammatory arthritis,…”, however, in light of numerous recent studies, authors should also mention the inflammatory component in OA.
➔ We added the inflammatory component in OA in current manuscript. Please see the highlight in yellow on line 141~146.
4. TNF-α should be replaced by only TNF.
➔ We replaced all TNF to TNF-α, because TNF-α would be more appropriate name for the molecule.
5. Authors should be careful about using the term “Animal pain” (line 205), it is more appropriate to use “animal pain behavior”.
➔ We corrected “animal pain” to “animal pain behavior” as your comment. You can see it in blue.
6. References 7, 89 need to be corrected.
➔ The previous reference number 7 and 89, you mentioned, were changed to 7,96 in the current manuscript, and I corrected them as your comment.
7. Authors should be uniform in using words “arthritis pain” or “arthritic pain”.
➔ The word “arthritic pain” was replaced to “arthritis pain” in the manuscript. You can see them in blue.
8. Line 352 reference is missing
➔ I added the reference on line 435 and highlighted in yellow.
9. Line 280, “animal arthritis” to “animal models of arthritis.
➔ We corrected it as your comment. Please see it in blue on line 360.
10. Line 336, “animal arthritis pain” to “animal models of arthritis pain”
➔ We corrected it as your comment. Please see it in blue on line 417.
11. Line 206,” experimental animal models” should be replaced either by “animal models” or “experimental models”
➔ We corrected it to “animal models”. Please see it in blue on line 271.
12. Line 335, Fibrocartilage, link should be removed.
➔ We removed the linkWe removed the link..